# Prevalence of hypertension, and related factors among adults in Wolaita, southern Ethiopia: A community-based cross-sectional study

**Wondimagegn Paulos Kumma**[1,2,3]*, **Bernt Lindtjørn**[1,2], **Eskindir Loha**[1,2,4]

**1** School of Public Health, Hawassa University, Hawassa, Ethiopia, **2** Centre for International Health, University of Bergen, Bergen, Norway, **3** School of Public Health, Wolaita Sodo University, Wolaita Sodo, Ethiopia, **4** Chr. Michelsen Institute, Bergen, Norway

* wondimagegnk@yahoo.com

## Abstract

### Introduction

Hypertension is a global public health challenge. There is a lack of evidence on the prevalence of hypertension, prehypertension, and related factors among adult populations of Wolaita, southern Ethiopia.

### Aim

To assess the prevalence of hypertension, prehypertension, and related factors among adult populations of Wolaita, southern Ethiopia.

### Methods

A community-based cross-sectional study was conducted on 2483 adult residents, selected using a two-stage random sampling technique. The quantitative data collected from structured questionnaires; anthropometric and biochemical measurements were entered into EpiData version 3.1 using double-entry systems. We determined the weighted prevalence of hypertension and pre-hypertension for the two-stage survey. The multivariate logistic regression analysis was used to assess factors associated with hypertension and carried out after declaring the data set as survey data to account for the effect of clustering. An adjusted coefficient with 95% CI was used to ascertain the significance of the association.

### Results

The weighted prevalence of hypertension and prehypertension in the Wolaita area was 31.3% (27.7%-35.1%) and 46.4% (42.9%-50.0%) respectively. The weighted prevalence of hypertension of those who were not aware of their hypertension until the time of the survey was 29.8%% (26.5%-33.3%). Where the weighted prevalence of self-reported cases of hypertension was 2.2% (1.2%-3.8%). Obesity, sugar-sweetened food consumption, male

**Data Availability Statement:** All relevant data are in the OSF repository and can be accessed at: https://osf.io/5wfah/ (DOI 10.17605/OSF.IO/5WFAH).

**Funding:** This study was funded by the South Ethiopia Network of Universities in Public Health (SENUPH); which in turn was funded by the Norwegian Program for Capacity Development in Higher Education and Research for Development (NORHED). The grant number of this research was ETH-13-0025. The URL of the funder is https://senuph.w.uib.no/. Wolaita Sodo University provided logistics and technical support. The funders had no role in study design, data collection and analysis, decision to publish, or preparation of the manuscript.

**Competing interests:** The authors have declared that no competing interests exist.

sex, elevated total cholesterol, raised fasting blood sugar, and advancing age were positively associated with hypertension.

## Conclusion

The prevalence of hypertension among adults in Wolaita was high. A small proportion of the affected people are aware of their high blood pressure. This study reported a high prevalence of pre-hypertension; which indicates a high percentage of people at risk of hypertension. It is essential to develop periodic screening programs, and primary intervention strategies such as the prevention of obesity, and reduction of sugar-sweetened food consumption.

## Introduction

The burden of non-communicable diseases (NCDs) in developing nations is increasing as a result of expanding urbanization, a growing economy, and shifting lifestyles [1, 2]. Hypertension, as a significant contributor to the burden of non-communicable diseases, is a global public health challenge [3, 4]. The global burden of hypertension is projected to be 1.56 billion in 2025; two-thirds of this will be occurring in developing countries [3]. Hypertension disproportionately affects populations in low- and middle-income countries [5].

Hypertension, defined as a systolic blood pressure greater or equal to 140 mmHg and/or diastolic blood pressure greater or equal to 90 mmHg or a self-reported case for medication [6]. Hypertension was believed to be rare in Africa, but it is currently perceived as one of the most important causes of cardiovascular diseases contributing to about 40% of the cases in the continent [5, 7]. According to the World Health Organization (WHO) report on NCDs 2014, hypertension among adults was highest in the African region with a prevalence of 30% [8]. It is a widespread problem with great economic impact because of its effect on productive subpopulations [9].

Studies around the world reported different levels of prevalence of hypertension and factors associated with blood pressure. A study conducted on the prevalence of hypertension among Indian adults reported an overall prevalence of 30% [10]. A study from Ghana showed a 13% prevalence of hypertension [11], and in Kenya, 24% of the population was hypertensive [12]. Factors reported as having associations with hypertension comprise older age, being male, being married, overweight, added sugar intake, alcohol drinking, and fruit consumption [10–14].

Evidence from the national NCDs STEPS survey of Ethiopia showed a 15% overall prevalence of hypertension [15]. Community-based cross-sectional surveys from other areas in Ethiopia reported prevalence rates ranging from 28% to 35% [16, 17]. In some places of southern Ethiopia, the prevalence of hypertension ranged from 22% to 35% [18, 19]. Identified risk factors of hypertension from studies in Ethiopia include older age, male sex, urban residence, higher formal education, physical inactivity, overweight or obesity, total cholesterol, raised fasting glucose, poor vegetable diet, and alcohol drinking [15–19]. In the study area, there is a lack of evidence on the prevalence and factors associated with hypertension (See S1 File). Therefore, this study aimed at determining the prevalence of hypertension, prehypertension, and related factors among adult populations of Wolaita, southern Ethiopia.

## Materials and methods

### Study design and setting

A community-based cross-sectional study was conducted from May 2018 to February 2019 in Wolaita Sodo town and Ofa rural areas, Wolaita Zone, southern Ethiopia. According to the projection based upon the 2007 population census, the population of Wolaita Zone in 2019 is about 2,042,593 people. Out of which, the proportion of the 25–64 years population is about 30.5%. We surveyed urban Wolaita Sodo and rural Ofa to assess variation in the prevalence of hypertension between urban and rural settings. Wolaita Sodo is the largest town in the zone with an estimated population of 165,596, and Ofa, one of the distant districts in the zone with a population of about 141,339 [20].

### Study subjects and sample size determination

The source population consists of people residing in the selected administrative areas of Wolaita Sodo town and Ofa rural district. The study population was permanent residents of randomly selected households aged between 25 to 64 years.

We calculated the sample size using Epi Info 7 StatCalc for two population proportions. Initially, the overall sample size (2486) was calculated to measure differences in dietary intake and nutrition transition among rural and urban populations as a general objective. Since our primary exposure variable is residence; the sample size was calculated based upon the prevalence of hypertension in urban and rural populations. The following assumptions from the study entitled prevalence of high blood pressure, hyperglycemia, dyslipidemia, metabolic syndrome and their determinants in Ethiopia: evidence from the national NCDs STEPS survey, 2015 were considered to calculate the sample size for this study [15]: 14.9% prevalence of hypertension in rural, 19.7% prevalence of hypertension in urban, 95% confidence level, 80% power, one for the ratio of unexposed and exposed groups, 10% non-response rate, and the total sample size became 2233. Therefore, the above sample size used in this study was sufficient to assess the prevalence of hypertension, prehypertension, and related factors among adult populations of Wolaita, southern Ethiopia.

### Sampling techniques and procedures

Data were collected in a two-stage cluster survey, with the villages from the urban and rural study sites being selected at the first stage, and a random sample of households within each village selected at the second stage. Initially, eleven out of 54 villages with urban characteristics of Wolaita Sodo town were selected using a simple random sampling technique. Similarly, ten out of 52 villages with rural characteristics of Ofa district were selected using a simple random sampling technique (Fig 1).

The distribution of samples in urban and rural areas was equal. Hence, 1243 people were selected from each study area. The estimated sample size was proportionally allocated to the selected villages based on their population size (Fig 1). The secondary unit of the study was the household and was selected using a simple random sampling technique from the list of enumerated households. To ensure the independence of observation, one eligible adult from each household was selected using simple random sampling. Each household was visited twice during the data collection time, one for the interview and anthropometric measurements, and the other for blood sample collection. After conducting interviews and taking anthropometric measurements during the first visit, the study participants were instructed overnight fasting and were appointed for blood sample collection for the next day.

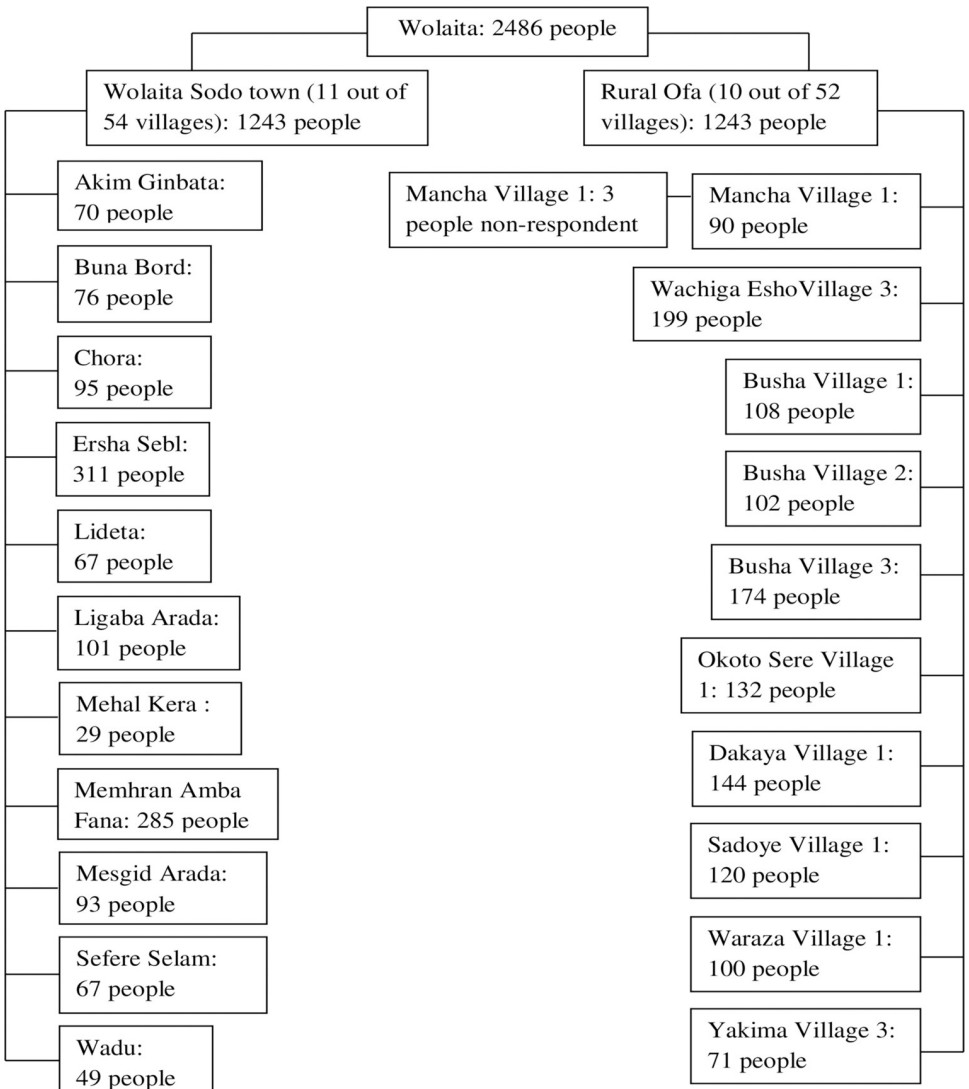

**Fig 1. Flow chart of the study subjects selection from villages in Wolaita Sodo town and rural Ofa, Wolaita, southern Ethiopia 2018.**

## Data collection procedures and techniques

Data for this study were collected using structured questionnaires, laboratory investigations, and anthropometric measurements. A series of questions about the potential risk factors and related variables were adapted from the World Health Organization (WHO) protocol for chronic non-communicable diseases (WHO STEPS survey) [21]. The English version (See S1 Questionnaire) of the questionnaire was translated into Amharic (See S2 Questionnaire), and Wolaita (See S3 Questionnaire) languages; and these were retranslated into English by an independent professional to ensure the accuracy of the translation.

Six field data collectors, five laboratory technicians, one field supervisor, one field coordinator, and two data clerks were recruited and given one week of training on data collection instruments. The training consisted of the purpose of the study, the contents of the questionnaire, interviewing skills, anthropometric measurements, laboratory procedures and analysis, format completion, and storage of samples.

Self-reported daily fruit and vegetable intakes were assessed using food frequency questions adapted from the WHO STEPwise approach to surveillance [21]. Physical activity was assessed based on the self-reported performance of moderate-intensity and vigorous-intensity activities, walking; time spent in minutes to carry out each activity, and MET (Metabolic equivalents) value of the respective activity. MET-minutes/week for a particular activity was computed by multiplying the number of days per week taken to perform each activity, with the time spent in minutes per day to perform the activity and the respective MET value of the activity [21, 22]. Finally, a combination of MET-minutes per week of walking, moderate-intensity, and vigorous-intensity activities was considered as the total MET-minutes/week [21, 22]. The recall period for the physical activity assessment was one week [21].

BMI was calculated as weight (kg) divided by height squared ($m^2$). Weight was measured to the nearest 0.1 kg using a portable digital weight scale (Seca electronic scale, 22089 Hamburg, Germany). The study subjects were weighted standing with light clothes on the scale with their shoes off. Before every measurement, the scale was tested for zero adjustments. Height was measured using a portable stadiometer (Seca, 22089 Hamburg, Germany), which consisted of a simple triangular headboard. For height measurement, the study subjects took off their shoes, stood straight, and held their head erect. The external auditory and the lower borders of the eyes were kept in one horizontal plane. The buttocks, shoulder blades, and heels touched the scale while legs with their knees stayed together and arms hanged by their sides. Height was measured to the nearest 0.1 cm. People with conditions not suitable for anthropometric measurements such as pregnant women and two people who were not suitable for height measurement such as participants who were unable to stand on the stadiometer were excluded from the study.

WHO STEPS data collection instrument was adapted for measurements of blood pressure (BP), and biochemical markers such as total cholesterol, fasting blood sugar level, and triglyceride values [21]. Whole venous blood samples were collected from participants in the morning after overnight fasting; and the application of 70% alcohol. Then the samples were stored in 3 ml vacutainer tubes holding ethylenediaminetetraacetic acid (EDTA). The test tubes with the samples were placed in the icebox and transported to Wolaita Sodo University (WSU) Hospital Laboratory for analysis of lipid profiles. Serum total cholesterol, HDL (High-Density Lipoprotein cholesterol), and triglycerides were determined using BS-200 Chemistry Analyzer with specific reagents for each biochemical value as per the manufacturer's instructions. The laboratory technicians performed the laboratory work within 12 hours of the blood sample collection at the WSU Hospital Laboratory. The fasting blood sugar level was determined on-site using a glucose meter (SensoCard®).

Blood pressure was measured using a digital sphygmomanometer (Riester, Germany). Blood pressure was measured three times, while the study subject was in a sitting position with the right upper arm placed at the level of the heart and after the subject had 10 minutes rest [23]. The average of the two measurements was considered to compute systolic and diastolic blood pressure [24]. There was a ten-minute interval between two blood pressure measurements.

## Operational definitions

**Hypertension.** Defined according to the 2018 European Society of Cardiology (ESC) and the European Society of Hypertension (ESH) Guidelines for the management of arterial hypertension (systolic blood pressure 140 mmHg or more and/or diastolic blood pressure 90 mmHg or more) and/ or self-reported for medication [6]. We classified hypertension based on the seventh report of the Joint National Committee, and the American College of Cardiology [25, 26].

Accordingly, systolic or diastolic blood pressure measurement < 120/80 mmHg is normal, 120-139/80-89 mmHg is pre-hypertension or elevated blood pressure, 140-159/90-99 mmHg is stage 1 hypertension, and ≥ 160/100 mmHg is stage 2 hypertension [25, 26].

**Hyperglycemia.**    Defined based upon the American Diabetes Association definition (persons with FBS level 7.0 mmol/l or above) and/ or self-reported for medication based on the prescription made by health personnel working in a licensed health institution [27].

**Nutritional status.**    BMI values of < 18.5 kg/m$^2$: underweight, 18.5–24.9 kg/m$^2$: normal, 25–29.9 kg/m$^2$: overweight and ≥ 30 kg/m$^2$: obese were used to classify the nutritional status of adults [28, 29].

**High total cholesterol level.**    Total cholesterol level 5.2 mmol/l (200 mg/dl) or more [30].

**Alcohol drinking.**    Defined based on the self-reported consumption of a standard alcoholic drink such as 285 ml of beer, 120 ml of wine, and 30 ml of spirits of five or more for men or four or more for women in a single drinking occasion within the past 30 days [21].

**Current smokers.**    Self-reported current use of smoked tobacco or smokeless tobacco products.

**Khat chewing.**    Defined based on the self-reported current chewing of khat.

**Low level of physical activity.**    If the study subject performed vigorous or moderate physical activity less than 600 METs-minutes/week [21, 22].

## Data quality management

Data collectors and immediate supervisors were trained for one week on the data collection instrument, key variables, and their measurements. A pre-test was conducted on 5% of the total sample size, with a population having the same socio-economic characteristics as the study population. The data collection team was offered retraining based on the problems identified and experiences gained during the pre-test. Laboratory technicians received training on standard operating procedures of blood sample collections. The principal investigator and field supervisors provided supportive supervision, daily throughout the data collection time. Incomplete and inconsistent data were returned to data collectors for corrections.

## Data entry and analysis

Data were entered into the Epi-Data version 3.1 (EpiData Association, Odense, Denmark) using the double-entry system and cleaned for inconsistency. The data were analyzed using STATA version 15 software (StataCorp LLC College Station Texas, U.S.A.). The wealth index was constructed using 40 variables for rural and 28 variables for urban areas related to the ownership of household assets using a principal component analysis. During the analysis, in each study setting eleven components with factor loading > 0.4 (house lighting, ceiling type, bedroom, cooking place, ox, electricity, radio, mobile, chair, table and mattress for a rural area, and floor type, wall type, ceiling type, bedroom, house ownership, toilet, television, mobile, table, mattress, and injera stove for an urban area) were identified and retained. The wealth index values were calculated by summing up the scores of eleven components in each study setting. Finally, the three socioeconomic categories were generated by splitting the wealth index values into three equal classes. Descriptive summaries were analyzed to determine the frequency and proportion of categorical variables. All the independent variables satisfied the assumption for the multicollinearity test with the Variance Inflation Factor (VIF) less than 10 or tolerance greater than 0.1.

Prevalence of pre-hypertension and hypertension was computed based on the European Society of Hypertension (ESH) Guideline [6]. To generate a weight for the two-stage cluster sampling, first, we computed finite population corrections for the two stages independently. Then to get our survey weight, we multiplied and inverted the two finite population

corrections. We determined the weighted prevalence, after declaring the data set as a two-stage survey, using the primary sampling unit identifier, the weight of the cluster, finite population correction of the first stage sampling, secondary sampling unit identifier, and finite population correction of the second stage sampling.

Factors associated with hypertension were analyzed using logistic regression. Binary categorical explanatory variables were coded in the same way as that of the outcome variable. Variables with P values less than 0.20 in the bivariate analysis, and those with socio-demographic and public health importance were selected for multivariate analysis. The number of cases that represent observation in the rarer of the two binary levels of the outcome variable was considered to fix the number of candidate variables that entered into the final multivariate logistic regression model. Accordingly, twelve variables were selected for multivariate analysis to control overfitting. The multivariate logistic regression analysis was started after declaring the data set as survey data to account for the effect of clustering in the estimated standard errors. AOR with 95% CI, and P value less than 0.05 were used to ascertain the significance of the association.

### Ethical consideration

Ethical clearance was obtained from the Institutional Review Board at Hawassa University (IRB/005/10) and the Regional Ethical Committee of Western Norway (2017/2248/REK nord). A support letter was obtained from Wolaita Sodo University and submitted to the concerned zonal and district offices. Written informed consent was obtained from all the study subjects; after introducing the purpose of the research using information sheets. Participation was based on voluntary. The identities of the study participants were kept confidential. Persons with FBS level 7.0 mmol/l (126 mg/dl) or above, high blood pressure, and other serious ailments were linked to the nearest health service facilities.

## Results

### Socio-demographic profiles of the participants

A total of 2483 respondents participated in the study with a response rate of 99.9%. The study participants were from two districts, namely Wolaita Sodo town (1243 (50.1%)) and Ofa rural district (1240 (49.9%)). The male to female ratio of the participants was 1.1. One thousand four hundred twenty-one (57.2%) study participants were between 25 and 39 years of age, with the median age of 35 ranging from 25–64 years. Of the total study participants, 676 (27.2%) were educated at the level of college or more. Most of the participants (2322 (93.5%)) were married or in relation (See Table 1).

### Prevalence of prehypertension and hypertension

The overall weighted prevalence of hypertension and prehypertension in the Wolaita area was 31.3% (27.7%-35.1%) and 46.4% (42.9%-50.0%) respectively. The weighted prevalence of hypertension of those who were not aware of their hypertension until the time of the survey was 29.8%% (26.5%-33.3%). Where the weighted prevalence of self-reported cases of hypertension was 2.2% (1.2%-3.8%). The weighted prevalence of hypertension in urban was 37.2% (28.3%-47.0%), while 28.7% (25.8%-31.8%) in the rural (See Table 1).

A remarkably higher weighted prevalence of hypertension was noted among participants with obesity (55.7% (45.3%-65.6%)) compared to the people with the normal nutritional status (29.6% (26.4%-33.0%)). The weighted prevalence of hypertension among participants who consumed sugar-sweetened food was 33.8% (29.5%-38.3%), while it was 23.6% (19.4%-28.4%) among people who did not consume sugar-sweetened food (See Table 2).

**Table 1. Distribution of socio-demographic and economic characteristics by hypertension among adults aged 25–64 years in Wolaita, southern Ethiopia 2018 (n = 2483).**

| Variables | Categories (n) | Prehypertension | | Hypertension | |
|---|---|---|---|---|---|
| | | Number of cases | Weighted prevalence % (95% CI) | Number of cases | Weighted prevalence % (95% CI) |
| **Age** | 25–39 years (1421) | 747 | 51.5 (46.7, 56.3) | 366 | 25.9 (21.9, 30.4) |
| | 40–64 years (1062) | 408 | 39.7 (35.6, 44.1) | 452 | 38.5 (33.3, 44.0) |
| **Sex** | Female (1170) | 552 | 46.5 (42.1, 50.8) | 340 | 28.8 (24.4, 33.6) |
| | Male (1313) | 603 | 46.4 (42.3, 50.6) | 478 | 33.4 (29.3, 37.9) |
| **Residence** | Rural (1240) | 575 | 46.3 (42.9, 50.0) | 356 | 28.7 (25.8, 31.8) |
| | Urban (1243) | 580 | 46.7 (38.3, 55.2) | 462 | 37.2 (28.3, 47.0) |
| **Educational status** | Primary & below (1410) | 611 | 44.6 (41.0, 48.2) | 445 | 29.9 (26.5, 33.6) |
| | High school (397) | 203 | 50.2 (45.0, 55.5) | 130 | 32.3 (27.1, 37.8) |
| | College+ (676) | 341 | 50.1 (40.7, 60.0) | 243 | 35.6 (26.7, 45.5) |
| **Ethnicity** | Wolaita (2372) | 1124 | 47.0 (43.5, 50.5) | 757 | 30.6 (27.3, 34.1) |
| | Others (111) | 31 | 27.6 (17.9, 40.0) | 61 | 55.5 (42.6, 67.7) |
| **Marital status** | Single (161) | 86 | 55.2 (48.6, 61.6) | 43 | 25.8 (20.5, 31.8) |
| | Married/ in relation (2322) | 1069 | 45.9 (42.1, 50.0) | 775 | 31.7 (28.0, 35.6) |
| **Occupation** | Employee (637) | 318 | 49.8 (39.1, 60.4) | 236 | 37.3 (27.3, 48.5) |
| | Merchant (332) | 164 | 49.4 (43.8, 55.1) | 102 | 29.4 (24.5, 34.9) |
| | Farmer (740) | 337 | 46.1 (41.6, 50.8) | 213 | 27.7 (24.7, 31.0) |
| | Housewife (509) | 219 | 42.9 (37.3, 48.7) | 160 | 31.0 (24.9, 37.8) |
| | Retiree (133) | 50 | 37.2 (29.6, 45.6) | 63 | 48.0 (40.7, 55.4) |
| | Students (74) | 43 | 55.7 (43.6, 67.2) | 17 | 24.8 (15.0, 38.2) |
| | Others (58) | 24 | 47.4 (33.7, 61.4) | 27 | 39.4 (24.4, 56.6) |
| **Wealth index** | Poor (784) | 321 | 42.4 (37.1, 47.8) | 288 | 34.4 (27.8, 41.7) |
| | Medium (793) | 364 | 46.8 (42.4, 51.2) | 263 | 30.5 (25.8, 35.6) |
| | Rich (906) | 470 | 49.8 (44.0, 55.5) | 267 | 29.3 (25.4, 33.4) |

Weighted prevalence: weighted for sampling.

## Factors associated with hypertension

Participants who developed obesity were significantly associated with hypertension [AOR = 2.5; 95% CI: 1.4–4.5] as compared to their counterparts. Similarly, there was a greater chance of developing hypertension among participants who consumed sweet food at least once a month [AOR = 1.6; 95% CI: 1.3–2.1] as compared to those who did not. Likewise, male participants had a greater chance of getting hypertension [AOR = 1.4; 95% CI: 1.1–1.7] compared to the female participants.

Moreover, hypertension increased with the increasing total cholesterol level [AOR = 1.2; 95% CI: 1.1–1.3] after adjusting the other factors. Hypertension was also increased with the increased blood sugar level [AOR = 1.1; 95% CI: 1.01–1.2] after adjusting the other factors. We observed a higher chance of developing hypertension with the advancing age [AOR = 1.03; 95% CI: 1.01–1.04] (Table 3).

## Discussion

The prevalence of hypertension among adults in Wolaita was high, with a weighted prevalence of 31.3%. A small proportion of the affected people are aware of their high blood pressure. This study reported a high prevalence of prehypertension; which indicates a high percentage of people at risk of hypertension. Obesity, sugar-sweetened food consumption, male sex,

**Table 2. Behavioral and anthropometric characteristics by hypertension among adults aged 25–64 years in Wolaita, southern Ethiopia 2018 (n = 2483).**

| Variables | Categories (n) | Prehypertension | | Hypertension | |
|---|---|---|---|---|---|
| | | Number of cases | Weighted prevalence % (95% CI) | Number of cases | Weighted prevalence % (95% CI) |
| **Physical activity** | Low (1311) | 626 | 46.5 (40.9, 52.2) | 451 | 34.5 (28.9, 40.5) |
| | Moderate/high (1172) | 529 | 46.4 (43.2, 49.6) | 367 | 28.8 (25.3, 32.5) |
| **Fruit consumption** | Not daily (60) | 22 | 36.3 (23.2, 51.8) | 21 | 33.0 (21.5, 47.0) |
| | Daily (2423) | 1133 | 46.6 (43.0, 50.2) | 797 | 31.3 (27.7, 35.1) |
| **Vegetable consumption** | Not daily (131) | 46 | 38.0 (24.3, 54.0) | 59 | 42.8 (30.6, 56.0) |
| | Daily (2352) | 1109 | 46.8 (43.1, 50.5) | 759 | 30.8 (27.3, 34.6) |
| **Sugar sweetened beverage intake** | No (1743) | 833 | 47.9 (44.4, 51.4) | 543 | 29.7 (26.3, 33.4) |
| | Yes (740) | 322 | 42.0 (35.0, 49.3) | 275 | 36.1 (29.8, 42.8) |
| **Sugar-sweetened food intake** | No (545) | 289 | 53.4 (48.4, 58.3) | 153 | 23.6 (19.4, 28.4) |
| | Yes (1938) | 866 | 44.2 (40.2, 48.3) | 685 | 33.8 (29.5, 38.3) |
| **Smoking** | No (2466) | 1150 | 46.6 (43.0, 50.2) | 812 | 31.4 (27.8, 35.2) |
| | Yes (18) | 5 | 29.2 (11.9, 55.8) | 6 | 24.8 (12.8, 42.5) |
| **Alcohol drinking** | No (2435) | 1142 | 46.7 (43.1, 50.4) | 790 | 31.0 (27.5, 34.7) |
| | Yes (48) | 13 | 29.3 (15.8, 47.9) | 28 | 50.4 (29.0, 71.7) |
| **BMI** | Underweight (411) | 170 | 41.2 (35.4, 47.3) | 117 | 27.9 (23.1, 33.2) |
| | Normal (1548) | 741 | 48.4 (45.4, 51.4) | 477 | 29.6 (26.4, 33.0) |
| | Overweight (415) | 208 | 48.6 (36.7, 60.7) | 163 | 41.2 (28.6, 55.1) |
| | Obese (109) | 36 | 31.6 (23.2, 41.3) | 61 | 55.7 (45.3, 65.6) |
| **Hyperglycemia** | No (2373) | 1122 | 47.1 (43.5, 50.7) | 757 | 30.6 (27.0, 34.3) |
| | Yes (110) | 33 | 29.5 (19.3, 42.2) | 61 | 50.6 (39.9, 61.3) |

Weighted prevalence: weighted for sampling.

elevated total cholesterol, raised fasting blood sugar, and advancing age were positively associated with hypertension.

The overall prevalence of hypertension reported in this study is in agreement with the findings reported from other studies [11, 13], while it is higher than the results indicated elsewhere in Ethiopia [14, 17], and other countries in Africa [12, 31]. The observed difference might be because of the socio-demographic and cultural variations among the study populations. The present study reported a higher prevalence of prehypertension compared to similar studies conducted in other African countries [11, 31]. It also reported a higher proportion of people who were not aware of their hypertension until the time of the survey compared to another study conducted in northwest Ethiopia [17]. This shows the presence of a high proportion of people that are at high risk of hypertension.

Participants who developed obesity were more likely to develop hypertension as compared to their counterparts. This is comparable with the findings from other studies [13, 16]. There is an established link between obesity and hypertension. The accumulation of excess fatty tissue instigates a cascade of events that give rise to increased blood pressure [32, 33]. Similarly, sugar-sweetened food consumption was also positively associated with hypertension, which is consistent with the finding from another study [14]. Excessive energy consumption might result in overweight and obesity and be linked to hypertension [34]. Furthermore, sex as a non-modifiable factor showed a positive association with hypertension, as has been found in another study [13, 17]. In this study, hypertension was more prevalent among male participants as compared to females.

**Table 3. A multivariate logistic regression using survey data analysis of factors with hypertension among the adult population aged 25–64 years in Wolaita, southern Ethiopia.**

| Variables (n = 2483) | Categories | Hypertension | | Crude OR (95% CI) | Adjusted OR (95% CI) |
|---|---|---|---|---|---|
| | | No N (%) | Yes N (%) | | |
| **Age** | | | | 1.03 (1.02, 1.05) | 1.03 (1.01, 1.04) |
| **Sex** | Female | 830 (70.9) | 340 (29.1) | 1.0 | 1.0 |
| | Male | 835 (63.6) | 478 (36.4) | 1.4 (1.1, 1.7) | 1.4 (1.1, 1.7) |
| **Residence** | Rural | 884 (71.3) | 356 (28.7) | 1.0 | 1.0 |
| | Urban | 781 (62.8) | 462 (37.2) | 1.5 (0.9, 2.4) | 1.2 (0.8, 1.9) |
| **Educational status** | Primary & below | 965 (68.4) | 445 (31.6) | 1.0 | 1.0 |
| | High school | 267 (67.2) | 130 (32.8) | 1.1 (0.8, 1.4) | 1.0 (0.7, 1.3) |
| | College+ | 433 (64.0) | 243 (36.0) | 1.2 (0.8, 1.9) | 0.9 (0.7, 1.2) |
| **Marital status** | Single | 118 (73.3) | 43 (26.7) | 1.0 | 1.0 |
| | Married/ in relation | 1547 (66.6) | 775 (33.4) | 1.4 (1.0, 1.9) | 1.0 (0.7, 1.4) |
| **Vegetable consumption** | Not daily | 72 (55.0) | 59 (45.0) | 1.0 | 1.0 |
| | Daily | 1593 (67.7) | 759 (32.3) | 0.6 (0.3, 0.9) | 0.6 (0.4, 1.1) |
| **Physical activity** | Low | 860 (65.6) | 451 (34.4) | 1.0 | 1.0 |
| | Normal | 805 (68.7) | 367 (31.3) | 0.9 (0.6, 1.2) | 1.0 (0.7, 1.4) |
| **Sugar sweetened food intake** | No | 412 (75.6) | 133 (25.4) | 1.0 | 1.0 |
| | Yes | 1253 (64.7) | 685 (35.4) | 1.7 (1.4, 2.1) | 1.6 (1.3, 2.1) |
| **Obesity** | No | 1617 (68.1) | 757 (31.9) | 1.0 | 1.0 |
| | Yes | 48 (44.0) | 61 (56.0) | 2.7 (1.7, 4.2) | 2.5 (1.4, 4.5) |
| **Alcohol drinking** | No | 1645 (67.6) | 790 (32.4) | 1.0 | 1.0 |
| | Yes | 20 (41.7) | 28 (58.3) | 2.9 (1.3, 6.7) | 2.1 (0.9, 4.9) |
| **Total cholesterol** | Mmol/ l | | | 1.3 (1.2, 1.4) | 1.2 (1.1, 1.3) |
| **Blood sugar** | Mmol/ l | | | 1.1 (1.1, 1.2) | 1.1 (1.01, 1.2) |

Moreover, hypertension increased with elevated total cholesterol levels. This is in line with the finding reported by another community-based study [15]. This might be due to increased deposition and accumulation of lipids in the blood vessels. Similarly, there was a positive association between fasting blood sugar level and hypertension. A study elsewhere reported a similar finding [17]. This might be partially explained by the sharing of common risk factors [35, 36]. Advancing age was associated with hypertension as has been found in other studies [16–18]. The decreasing energy expenditure with advancing age may lead to the accumulation of adipose tissue and this may give rise to the development of obesity and high blood pressure [32, 37].

## Strengths and limitations

The magnitude of confounding was assessed using the level of variation between the crude and adjusted estimates. The absence of difference between the two estimates indicates the observed exposure-outcome effect was not confounded by the potential confounding variable. The study, being a cross-sectional survey, lacks a temporal relationship. Since Wolaita is a predominantly religious society, we expect a social desirability bias for responses related to behavioral questions such as smoking, alcohol, and khat chewing. In this study, the distribution of sex across the age groups was not as expected. Age was measured based on the birth date estimation using the study participants' recall memory, which was supported by main public events that occurred around the participants' birth date.

## Conclusion

The prevalence of hypertension among adults in Wolaita was high. A small proportion of the affected people are aware of their high blood pressure. This study reported a high prevalence of prehypertension; which indicates a high percentage of people at risk of hypertension. Obesity, sugar-sweetened food consumption, male sex, elevated total cholesterol level, raised fasting blood sugar level, and advancing age were positively associated with hypertension. We identified modifiable risk factors with public health importance that includes obesity, sugar-sweetened food consumption, elevated total cholesterol, and raised fasting blood sugar level. The findings of this study can be used for immediate public health practice. Therefore, it is essential to develop periodic screening programs, and primary intervention strategies such as the prevention of obesity, and reduction of sugar-sweetened food consumption as has been shown in this study.

## Supporting information

**S1 File. Knowledge gap on factors associated with systolic and diastolic blood pressures and comparison of mean systolic and diastolic blood pressures considering residence as a primary exposure variable in Ethiopia.**
(RAR)

**S1 Questionnaire. English questionnaire and consent.**
(RAR)

**S2 Questionnaire. Amharic questionnaire and consent.**
(RAR)

**S3 Questionnaire. Wolaita language questionnaire version and consent.**
(RAR)

## Acknowledgments

We would like to thank the Norwegian Program for Capacity Development in Higher Education and Research for Development (NORHED). We would like to acknowledge Wolaita Sodo University for logistics and technical support. The authorities in the districts and zone are also thanked. We are grateful to the cooperation of the study participants and data collectors, for taking part in this work.

## Author Contributions

**Conceptualization:** Wondimagegn Paulos Kumma, Bernt Lindtjørn, Eskindir Loha.

**Data curation:** Wondimagegn Paulos Kumma, Bernt Lindtjørn, Eskindir Loha.

**Formal analysis:** Wondimagegn Paulos Kumma, Bernt Lindtjørn, Eskindir Loha.

**Funding acquisition:** Wondimagegn Paulos Kumma, Bernt Lindtjørn, Eskindir Loha.

**Investigation:** Wondimagegn Paulos Kumma, Bernt Lindtjørn, Eskindir Loha.

**Methodology:** Wondimagegn Paulos Kumma, Bernt Lindtjørn, Eskindir Loha.

**Project administration:** Wondimagegn Paulos Kumma, Bernt Lindtjørn, Eskindir Loha.

**Resources:** Wondimagegn Paulos Kumma, Bernt Lindtjørn, Eskindir Loha.

**Software:** Wondimagegn Paulos Kumma.

**Supervision:** Wondimagegn Paulos Kumma, Bernt Lindtjørn, Eskindir Loha.

**Validation:** Wondimagegn Paulos Kumma, Bernt Lindtjørn, Eskindir Loha.

**Visualization:** Wondimagegn Paulos Kumma, Bernt Lindtjørn, Eskindir Loha.

**Writing – original draft:** Wondimagegn Paulos Kumma.

**Writing – review & editing:** Wondimagegn Paulos Kumma, Bernt Lindtjørn, Eskindir Loha.

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
