## [Decision Letter · Decision Letter 0]

16 Apr 2021

PONE-D-21-02507

Hypertension in southern Ethiopia: a community-based cross-sectional study from in Wolaita

PLOS ONE

Dear Dr. Kumma,

Thank you for submitting your manuscript to PLOS ONE. After careful consideration, we feel that it has merit but does not fully meet PLOS ONE’s publication criteria as it currently stands. Therefore, we invite you to submit a revised version of the manuscript that addresses the points raised during the review process.

The manuscript has been evaluated by two reviewers, and their comments are available below. The reviewers have raised a number of concerns that need attention. They request additional information on methodological aspects of the study and the interpretation of the results. Please pay particular attention to the reviewers' requests to clarify the main research question of your study, i.e. the prevalence of hypertension versus systolic/diastolic blood pressure. Could you please revise the manuscript to carefully address the concerns raised?

We look forward to receiving your revised manuscript.

Kind regards,

Dario Ummarino, Ph.D.

Senior Editor

PLOS ONE

Journal Requirements:

"No: The funders had no role in study design, data collection and analysis, decision to publish, or preparation of the manuscript."

Reviewers' comments:

Reviewer's Responses to Questions

**Comments to the Author**

1. Is the manuscript technically sound, and do the data support the conclusions?

Reviewer #1: Partly

Reviewer #2: Yes

2. Has the statistical analysis been performed appropriately and rigorously? 

Reviewer #1: No

Reviewer #2: I Don't Know

3. Have the authors made all data underlying the findings in their manuscript fully available?

Reviewer #1: Yes

Reviewer #2: No

4. Is the manuscript presented in an intelligible fashion and written in standard English?

Reviewer #1: Yes

Reviewer #2: Yes

5. Review Comments to the Author

Reviewer #1: Thank you very much for giving me opportunity to revise this paper.

Hypertension in southern Ethiopia: a community-based cross-sectional study from in

Wolaita

The title, introduction, discussion are all on hypertension straight away. However, the objective and statistics, methods and results are on the level of systolic and diastolic blood pressure. This is confusing because these are two different issues. Thus the paper is on tow different objectives. Authors wrongly cited articles on hypertension as if these are on systolic and diastolic blood pressure. I have gone through your references, non of them is on systolic and diastolic blood pressure.

Reviewer #2: Title ----I think community based cross-sectional study is enough delete from in Wolita

Abstract: objective

Line 23---if your aim was to assess factors associated with systolic and diastolic blood pressure, best study design is case control but you use cross-sectional why?

--You assessed systolic and diastolic blood pressure or systolic and diastolic high blood pressure?

Line 32---In your result you start about prevalence of hypertension, but your objective was systolic and diastolic blood pressure, if it is not about high blood pressure how you could talk about hypertension??? Is this consistent?

Line/no 33-34----your mean systolic BP was 130.5 with CI of 129.8-131.3mmHg; is this hypertension?? Because as my understanding and as your category this is under prehypertension. What you say on this??

You calculate mean systolic and diastolic blood pressure from over all but it is good if you calculate from those above 140(systolic) or high blood pressure and similarly to diastolic above 90.

Your document lacks consistency ---for example in objective you said to compare systolic and diastolic blood pressure among urban and rural but in your method section you said to assess variation of prevalence of hypertension among urban and rural. Which one is your focus?

154---Data collection----narrating all issue is I think boring, so try to minimize it

186------not suitable for ht measurement was excluded—what is not suitable and suitable??? Please clearly mention it.

157---I think you adapt your survey questions from WHO steps survey not prepared, are you prepared or adapt?

Result section—

Mean systolic and mean diastolic blood pressure—what is the importance of assessing the mean systolic and diastolic blood pressure? What is the public health importance of the mean value of systolic and diastolic blood pressure?

337---Those not aware of their hypertension were 30.9% and self-report was 3%: according to your operational definition it was 33.9% but yours is 32.9%; what makes this difference?

Line 317---table 3 demonstrates…..participants.—rather it is good you simply write issues you put under table 3 and at last or end of paragraph refer to table 3 or put in bracket like(see table 3)

Line 318-320---In your operational definition you put normal value and what hyper…means. I think it is not necessary to write this issue in this part.

Almost in all your result you include or write exact value of participant and total sample size. Example in line 321 589 0f 2483; but the sample size or your study participant are known or you tell us at the beginning of your result so it is good you put number and percent with out your total sample size or participant.

-line 329-331

Factors associated with…… and table 4 and 5 heading is not shows us associated factors please revise it.

Line 356 or table 4 you show us distribution of systolic and diastolic blood pressure or hypertension and in urban ---for stage I HTN=9.7 and stage II=6.8 totally ===16.5

For rural---stage I=10.2 and stage II=4.2 totally===14.4 according to your operational definition prevalence of hypertension is 16.5% newly diagnosed plus self report 5.2=21.7% for urban and 14.4% new plus 0.8% self report =15.2% for rural. But your prevalence is different from this, how you calculate this? Are you include pre-hypertension?? If so is this right?

Additionally you reported self report 3% but in your table it is 6%(0.8% rural and 5.2% urban) what makes this difference?

Again on residence variable both urban and rural row there is summation error please see and revise it.

Line 374---Similar comment as line 317

Starting from line 396----- you tell us about factors associated with systolic as well as diastolic blood pressure that is including normal blood pressure, so what is public health importance and what you recommend based on this??

Factors associated with mean systolic and mean diastolic blood pressure including normal why?

I didn’t see any discussion issue about isolated systolic hypertension and isolated diastolic hypertension why?

Table 7 ---editorial issue, it is mean diastolic blood pressure

Strength and limitation

Starting from line 509----please try to minimize it. Avoid unnecessary issue, some you included under strength are not strength and it is your sampling technique and others are about method, so please revise it.

Line 534---advancing age or advanced age

Finally---your title is about Hypertension in southern Ethiopia, your objective is to assess factors associated with systolic and diastolic blood pressure and compare the mean systolic and diastolic blood pressures among rural and urban populations and you conclude as prevalence of hypertension in Wolita was high. How you see these?? Your objective was not about high blood pressure but simply systolic and diastolic blood pressure and through out your document you tell us about this issue, so how it could be about hypertension??

6. PLOS authors have the option to publish the peer review history of their article (what does this mean?). If published, this will include your full peer review and any attached files.

Reviewer #1: No

Reviewer #2: No

---

## [Author Response · Author response to Decision Letter 0]

19 May 2021

PONE-D-21-02507

Hypertension in southern Ethiopia: a community-based cross-sectional study 

Dear Dr. Dario Ummarino,

We would like to thank you and the reviewers for considering our paper for the next level. We have carefully worked on the comments provided by the academic editor as well as the reviewers. We have uploaded the following items during our submission: responses to the reviewers, Revised Manuscript with Track Changes, and Revised Paper without Track Changes (Manuscript). Please see below our one by one responses and explanations to questions and comments. Our responses and explanations to the questions and comments are indicated in bullet points. 

• Thank you so much, we have worked according to the PLOS ONE requirement. 

 2. Thank you for stating the following financial disclosure: "No: The funders had no role in study design, data collection and analysis, decision to publish, or preparation of the manuscript."

a. Please clarify the sources of funding (financial or material support) for your study. List the grants or organizations that supported your study, including funding received from your institution.

• South Ethiopia Network of Universities in Public Health (SENUPH) funded the research; which in turn is funded by the Norwegian Program for Capacity Development in Higher Education and Research for Development (NORHED). 

• Wolaita Sodo University provided logistics and technical support.

b. State what role the funders took on the study. If the funders had no role in your study, please state: “

• The funders had no role in study design, data collection, and analysis, decision to publish, or preparation of the manuscript.

• The principal author is the academic staff of Wolaita Sodo University, Ethiopia. Therefore, I receive a salary from the Wolaita Sodo University, Ethiopia. However, the university had no role in the study design, data collection and analysis, decision to publish, or preparation of the manuscript. Currently, I am on study leave for a PhD study. I am doing my PhD study in a joint PhD program at Hawassa University and the University of Bergen. And this study is part of my PhD work, where the Wolaita Sodo University has no direct role in this study.

d. If you did not receive any funding for this study, please state: “The authors received no specific funding for this work.”

• The authors received no specific funding for this work. 

• Thank you, we will provide the relevant accession numbers or DOIs necessary to access our data.

Comments to the Author

1. Is the manuscript technically sound, and do the data support the conclusions?

Reviewer #1: Partly

Reviewer #2: Yes

• Thank you for the feedback; we have used all our effort to satisfy the reviewers regarding the comment.

2. Has the statistical analysis been performed appropriately and rigorously?

Reviewer #1: No

Reviewer #2: I Don't Know

• Thank you for the feedback, we have worked to answer the comments provided by the reviewers. Depending upon the comments provided by the academic editor and the reviewers; we have modified the title, and objective of the study. We have also worked on the introduction and discussion section based upon the modifications made to the title and the objective. Primarily, we did a linear regression analysis to assess factors associated with systolic and diastolic blood pressure in Wolaita, southern Ethiopia. In line with the modifications made to the title and the objective, we added some points in the statistical analysis to address the prevalence of hypertension.

3. Have the authors made all data underlying the findings in their manuscript fully available?

Reviewer #1: Yes

Reviewer #2: No

• Thank you for the comment; we will deposit the data to a public repository on the acceptance of the manuscript for publication.

4. Is the manuscript presented in an intelligible fashion and written in standard English?

Reviewer #1: Yes

Reviewer #2: Yes

• Thank you for the feedback.

5. Review Comments to the Author

Reviewer #1: Thank you very much for giving me opportunity to revise this paper.

Hypertension in southern Ethiopia: a community-based cross-sectional study from in

Wolaita. The title, introduction, discussion are all on hypertension straight away. However, the objective and statistics, methods and results are on the level of systolic and diastolic blood pressure. This is confusing because these are two different issues. Thus the paper is on two different objectives. Authors wrongly cited articles on hypertension as if these are on systolic and diastolic blood pressure. I have gone through your references, none of them is on systolic and diastolic blood pressure.

• Thank you, we accepted the comment. Accordingly, we have modified the title, and objective of the study. That is, the title now has both hypertension and blood pressure. And in line with it, we have incorporated additional points in the introduction and discussion section of the paper. We have also used additional literatures while making changes to the introduction and discussing our findings. Therefore, the reference section of our paper is reorganized in line with the modifications.

Reviewer #2: Title ----I think community based cross-sectional study is enough delete from in Wolaita

• Thank you, we have accepted the comment, and rewrote the title as suggested.

Abstract: objective

Line 23---if your aim was to assess factors associated with systolic and diastolic blood pressure, best study design is case control but you use cross-sectional why?

• We appreciate your concern. Our explanation to the question is, it is possible to use either of the two designs depending upon the research question or the purpose of the study. In retrospective studies, which start with the case (disease/ outcome) and then possible exposure, a case-control study design is a commonly used study design. Whereas, a cross-sectional study design can also be used to assess the association of different factors with an outcome variable for the data collected at a point in time. In our case, we collected primary data on blood pressure and other variables at a point in time, which is cross-sectional. And we are aware that the temporality of the association between the exposure and the outcome variable remains in question except for some biological exposures like sex. We took this into consideration while interpreting the results. 

You assessed systolic and diastolic blood pressure or systolic and diastolic high blood pressure?

• We assessed systolic and diastolic blood pressure, and we categorized according to the operational definition of high blood pressure for the sake of reporting the burden of hypertension.

Line 32---In your result you start about prevalence of hypertension, but your objective was systolic and diastolic blood pressure, if it is not about high blood pressure how you could talk about hypertension??? Is this consistent?

• Thank you for raising such an important point, which is similar to the question raised by the first reviewer. As we responded to the question raised by the first reviewer; taking our important findings and consistency of information starting from the title to discussion into consideration, we have modified our title, and the objective. Accordingly, we have also incorporated additional points to the introduction, methods, and discussion. In line with it, we have used additional literatures.

Line/no 33-34----your mean systolic BP was 130.5 with CI of 129.8-131.3mmHg; is this hypertension?? Because, as my understanding and as your category this is under prehypertension. What you say on this??

• Thank you for the question and for providing this opportunity to clarify our research work. As you know blood pressure data is continuous data. Therefore, it is possible to compute the mean with its confidence interval from the overall systolic as well diastolic blood pressure data. The mean does not provide categorical information; it gives aggregate information on the average blood pressure within the study population. 

In order to address the prevalence of pre-hypertension and hypertension, we have modified our title and objective. And analyzed the blood pressure data by classifying it into different categories according to the European Society of Hypertension (ESH) Guidelines. Using this we presented results on pre-hypertension and hypertension with its stages. 

You calculate mean systolic and diastolic blood pressure from overall, but it is good if you calculate from those above 140(systolic) or high blood pressure and similarly to diastolic above 90.

• Thanks for this question also. One of the advantages of continuous data analysis of the overall observations is to avoid loss of information. An important property of the mean is that it includes every value in your data set as part of the calculation, and its purpose is to indicate the central location of our data since the mean is a measure of central tendency. If we start to compute the mean of systolic and diastolic blood pressure from above 140 and 90 mmHg, respectively, then it will be the mean of blood pressure for the hypertensive category only and may not be as such valuable information considering the overall aim or context of our study. Because of this reason we computed mean from the overall data. 

Your document lacks consistency ---for example in objective you said to compare systolic and diastolic blood pressure among urban and rural but in your method section you said to assess variation of prevalence of hypertension among urban and rural. Which one is your focus?

• Thank you so much for indicating such a discrepancy. We have made corrections based upon your comment. Please, see the correction in the revised paper without track change in line number 130, and 131. 

154---Data collection----narrating all issue is I think boring, so try to minimize it

• Thank you, we have deleted some waste words from the document. Please, see deletions in the revised paper with track change in line number 146, 147, 160, and 277.

186------not suitable for ht measurement was excluded—what is not suitable and suitable??? Please clearly mention it.

• Thank you so much, we have now mentioned the condition which was not suitable for height measurement. Please, see the correction in the revised paper without track change in line number 188.

157---I think you adapt your survey questions from WHO steps survey not prepared, are you prepared or adapt?

• Thank you so much for the comment. We have paraphrased the statement as per the comment. Please, see the correction in the revised paper without track change in line number 157.

Result section—

Mean systolic and mean diastolic blood pressure—what is the importance of assessing the mean systolic and diastolic blood pressure? What is the public health importance of the mean value of systolic and diastolic blood pressure?

• Thank you for raising such an important concern. We think that computing the mean values provides the overall picture of blood pressure in the studied community without any loss of information due to groupings into high and low. Meanwhile, we have also provided categorical information on levels of blood pressure, including the magnitude of hypertension considering the public health importance of such information. And we believe that reporting in both ways is an advantage.

337---Those not aware of their hypertension were 30.9% and self-report was 3%: according to your operational definition it was 33.9% but yours is 32.9%; what makes this difference?

• Thank you so much for allowing us to explain this, as you saw in the text in line 337, the prevalence of hypertension of those who were not aware of their hypertension until the time of the survey, meaning the prevalence of undiagnosed hypertension was 30.9% (743 of 2408 persons; 95% CI: 29.0% - 32.7%). Here the prevalence of hypertension was calculated from the newly diagnosed participants meaning we excluded the self-reported cases who had an awareness of their previous hypertension status from the analysis. That means the denominator, in this case, is 2408 as indicated in the text. Whereas, the denominator for the self-reported cases of hypertension was 2483 (3.0% (75 of 2483 persons; 95% CI: 2.4%-3.8%)). Because of the difference in the denominator, we did not add the two proportions. Taking 2483 as the denominator for both, it is possible to obtain 32.9%. There is also an overlap of hypertension cases between previously undiagnosed cases and self-reported cases of medication as shown in Table 1, which is prepared for the sake of clarifying this question.

Table 1. Distribution of hypertension with high blood pressure (high systolic and diastolic blood pressures) and self-reported cases of medication, Wolaita, Southern Ethiopia (n = 2483). 

Category Self-reported cases Total 

 No yes 

HBP No 0 (0.0) 16 (0.6) 16 (0.6)

 Yes 743 (29.9) 59 (2.4) 802 (32.3)

Total 743 (29.9) 75 (3.0) 818 (32.9)

Line 317---table 3 demonstrates…..participants.—rather it is good you simply write issues you put under table 3 and at last or end of paragraph refer to table 3 or put in bracket like(see table 3)

• Thank you, we accepted your comment and corrected it accordingly. Please, see such types of corrections in the revised paper without track change in line number 317, 328, 336, 356, 361, 399, 405, 412, 424, 432, and we have omitted the statements mentioned at the beginning of the paragraphs.

Line 318-320---In your operational definition you put normal value and what hyper…means. I think it is not necessary to write this issue in this part.

• We also thank you for this comment, we have deleted the redundancies. Please, see the changes in the revised paper without track change in line number 332, and 333.

Almost in all your result you include or write exact value of participant and total sample size. Example in line 321 589 0f 2483; but the sample size or your study participant are known or you tell us at the beginning of your result so it is good you put number and percent with out your total sample size or participant.

• Thank you, we have made corrections on it. Please, see an example of the corrections in the revised paper without track change from line number 332 to 336. Accordingly, we have made such types of corrections throughout the paper, except for conditions where there is a difference in the denominator as indicated in line number 345.

-line 329-331 Factors associated with…… and table 4 and 5 heading is not shows us associated factors please revise it.

• Thanks, we have rewritten the headings of Tables 4 and 5. Please, see the corrections in the revised paper without track change in line number 362, 363, 366, and 367.

Line 356 or table 4 you show us distribution of systolic and diastolic blood pressure or hypertension and in urban ---for stage I HTN=9.7 and stage II=6.8 totally ===16.5

For rural---stage I=10.2 and stage II=4.2 totally===14.4 according to your operational definition prevalence of hypertension is 16.5% newly diagnosed plus self report 5.2=21.7% for urban and 14.4% new plus 0.8% self report =15.2% for rural. But your prevalence is different from this, how you calculate this? Are you include pre-hypertension?? If so is this right?

• We appreciate your comment on Table 4. Based upon your comment we have revised table four, and made some changes on the denominators. Now, the denominator became the row total for all the categories to simplify comparison between categories of variables. Because of the difference in the denominator of the two categories (Column 8 & 9), the addition of prevalence rates cannot provide the same result. We did not mix pre-hypertension with hypertension cases. Please, see the changes made in the revised paper without track change on line number 364 or Table 4.

Additionally you reported self report 3% but in your table it is 6% (0.8% rural and 5.2% urban) what makes this difference?

• Thank you, as we have explained above the denominators are different, therefore the summation does not provide 3%. Please, see the revised paper without track change in line number 364 or Table 4.

Again on residence variable both urban and rural, row there is summation error please see and revise it.

• Thank you for the comment, we have cross-checked the summation. The row total for rural and urban excluding the rows in column 7, table 4 is 1240 and 1243 respectively. The results in column 7 are the summation of row results from columns 1 - 6. Please, see the revised paper without track change in line number 364 or Table 4.

Line 374---Similar comment as line 317

• Thank you, we have accepted the comment and made the correction accordingly. Please, see the correction in the revised paper without track change in line number 399, and we have omitted the statement which was written at the beginning of the paragraph.

Starting from line 396----- you tell us about factors associated with systolic as well as diastolic blood pressure that is including normal blood pressure, so what is public health importance and what you recommend based on this??

• Thank you so much for the question. Our explanation of this question is similar to other questions related to it. Systolic and diastolic blood pressure data are continuous data. We can analyze systolic and diastolic blood pressure as the dependent variables using linear regression, which does not require dropping values or systematically categorize them. The advantage of analyzing such data using linear regression is to identify factors that contribute either to the increase or decrease of a blood pressure level in the study population. Therefore, using the information obtained from this study public health measures can be taken to control modifiable risk factors that contribute to the increase of blood pressure, and factors decreasing blood pressure in the community can be promoted. Analyzing continuous data using linear regression avoids loss of information. 

Factors associated with mean systolic and mean diastolic blood pressure including normal why?

• We analyzed factors associated with systolic and diastolic blood pressure. Our explanation is, with linear regression using the overall data; one can observe the relationship between the explanatory and an outcome variable. Whether there is a linear increase in the outcome variable with the increasing explanatory variable or a linear decrease in the outcome variable with the decreasing explanatory variable.

I didn’t see any discussion issue about isolated systolic hypertension and isolated diastolic hypertension why?

• Thank you, we have now included it in the discussion. Please, see the discussion made on the isolated systolic, and diastolic hypertension in the revised paper without track change from line number 456 to 462.

Table 7 ---editorial issue, it is mean diastolic blood pressure

• Thank you; it is diastolic blood pressure.

Strength and limitation

Starting from line 509----please try to minimize it. Avoid unnecessary issue, some you included under strength are not strength and it is your sampling technique and others are about method, so please revise it.

• Thank you so much, we have revised it and avoided issues related to methods. Please, see the corrections in the revised paper with track changes from line number 539 to 543.

Line 534---advancing age or advanced age

• Thank you for raising the point, some use advancing age others use advanced age. However, we used the term "advancing" to show the increasing trend or pattern of blood pressure with increasing age. 

Finally---your title is about Hypertension in southern Ethiopia, your objective is to assess factors associated with systolic and diastolic blood pressure and compare the mean systolic and diastolic blood pressures among rural and urban populations and you conclude as prevalence of hypertension in Wolita was high. How you see these?? Your objective was not about high blood pressure but simply systolic and diastolic blood pressure and throughout your document you tell us about this issue, so how it could be about hypertension??

• Thank you so much for showing us this important point. Taking your comments into consideration, we have modified our objective and in line with it, we have modified our title, and incorporated additional points into the introduction of the paper. We have also incorporated additional points to the methods and discussion, and used additional articles that address the modified objective. 

6. PLOS authors have the option to publish the peer review history of their article (what does this mean?). If published, this will include your full peer review and any attached files.

• No 

Do you want your identity to be public for this peer review? For information about this choice, including consent withdrawal, please see our Privacy Policy.

Reviewer #1: No

Reviewer #2: No

---

## [Decision Letter · Decision Letter 1]

29 Jun 2021

PONE-D-21-02507R1

Hypertension, and blood pressure in southern Ethiopia: a community-based cross-sectional study

PLOS ONE

Dear authors/editorial staff members/Editor in chief! 

Thank you for submitting your manuscript to PLOS ONE. After careful consideration, we feel that it has merit but does not fully meet PLOS ONE’s publication criteria as it currently stands. Therefore, we invite you to submit a revised version of the manuscript that addresses the points raised during the review process.

We look forward to receiving your revised manuscript.

Kind regards,

Wali Khan

Academic Editor

PLOS ONE

Journal Requirements:

Additional Editor Comments (if provided):

Dear authors/editorial staff members/Editor in chief!

I would like to inform you that I have reached my decision that the manuscript numbered cited in the subject above is based on scientific background and fulfil and satisfy the standard of PLOS ONE for publication. I have checked out the points of reviewers raised during revision and responses given by the author(s) against each point. I am satisfied by the response of the author(s) for the points raised. As according to the reviewers comments objectives were not similar to the title and thus the findings were not related to the objectives and lack of consistency in the paper but this is now addressed by the authors accordingly. As Academic editor of this manuscript I decided to publish this submission in PLOS ONE.

Reviewers' comments:

Reviewer's Responses to Questions

**Comments to the Author**

1. If the authors have adequately addressed your comments raised in a previous round of review and you feel that this manuscript is now acceptable for publication, you may indicate that here to bypass the “Comments to the Author” section, enter your conflict of interest statement in the “Confidential to Editor” section, and submit your "Accept" recommendation.

Reviewer #1: All comments have been addressed

Reviewer #2: All comments have been addressed

2. Is the manuscript technically sound, and do the data support the conclusions?

Reviewer #1: Yes

Reviewer #2: Yes

3. Has the statistical analysis been performed appropriately and rigorously? 

Reviewer #1: Yes

Reviewer #2: I Don't Know

4. Have the authors made all data underlying the findings in their manuscript fully available?

Reviewer #1: Yes

Reviewer #2: No

5. Is the manuscript presented in an intelligible fashion and written in standard English?

Reviewer #1: Yes

Reviewer #2: Yes

6. Review Comments to the Author

Reviewer #1: I suggest to accept it in the current form.

Authors succeeded to response to all of the raised points

Reviewer #2: Minor comment

Result

Line 371---mean(SD) it is good if you write like 130.5 ±0.4

Discussion

Line 476-478---Why you raised this issue? Because as you told us this was not assed in your study. So it is good if you delete it.

Strength

Line 509---judgment of causality?? Your study design is cross-sectional how it could be???

As my perception it is good you minimize or summarize it and make your strength & limitation to one paragraph

Conclusion

Line 536—you write modifiable risk factors but it is good if you indicate this statement above (line 529) when you listing factors associated with blood pressure. Because this is conclusion so it is good if you show us those risk factors you identified in your study as modifiable and non-modifiable.

Reference

Line 596—reference 10—randomly I checked this reference and for this article there is authors so, it is good if you see and revise it.

7. PLOS authors have the option to publish the peer review history of their article (what does this mean?). If published, this will include your full peer review and any attached files.

Reviewer #1: **Yes: **Ishag Adam

Reviewer #2: No

---

## [Author Response · Author response to Decision Letter 1]

2 Jul 2021

Dear Dr. Wali Khan,

We would like to thank you and the reviewers for your suggestions, and comments. We have worked on the comments provided by the reviewer. We have uploaded the following items during our resubmission: responses to the reviewers, Revised Manuscript with Track Changes, and Revised Paper without Track Changes (Manuscript). Please see below our one-by-one response and explanations to questions and comments. Our responses and explanations to the questions and comments are indicated in bullet points. 

Journal Requirements:

• Thank you so much, we have carefully reviewed the reference list, and all are written according to the PLOS ONE requirement.

Additional Editor Comments (if provided):

Dear authors/editorial staff members/Editor in chief!

I would like to inform you that I have reached my decision that the manuscript numbered cited in the subject above is based on scientific background and fulfil and satisfy the standard of PLOS ONE for publication. I have checked out the points of reviewers raised during revision and responses given by the author(s) against each point. I am satisfied by the response of the author(s) for the points raised. As according to the reviewers comments objectives were not similar to the title and thus the findings were not related to the objectives and lack of consistency in the paper but this is now addressed by the authors accordingly. As Academic editor of this manuscript I decided to publish this submission in PLOS ONE.

• Thank you so much for such an encouraging response.

Reviewers' comments: Reviewer's Responses to Questions

Comments to the Author

1. If the authors have adequately addressed your comments raised in a previous round of review and you feel that this manuscript is now acceptable for publication, you may indicate that here to bypass the “Comments to the Author” section, enter your conflict of interest statement in the “Confidential to Editor” section, and submit your "Accept" recommendation.

Reviewer #1: All comments have been addressed

Reviewer #2: All comments have been addressed

• Thank you so much.

2. Is the manuscript technically sound, and do the data support the conclusions?

Reviewer #1: Yes

Reviewer #2: Yes

• Thank you so much.

3. Has the statistical analysis been performed appropriately and rigorously?

Reviewer #1: Yes

Reviewer #2: I Don't Know

• Thank you so much.

4. Have the authors made all data underlying the findings in their manuscript fully available?

Reviewer #1: Yes

Reviewer #2: No

• Thank you for the comment; as we have responded previously, we will deposit the data to a public repository on the acceptance of the manuscript for publication.

5. Is the manuscript presented in an intelligible fashion and written in standard English?

Reviewer #1: Yes

Reviewer #2: Yes

• Thank you so much.

6. Review Comments to the Author

Reviewer #1: I suggest to accept it in the current form.

Authors succeeded to response to all of the raised points

• Thank you so much.

Reviewer #2: Minor comment

Result

Line 371---mean(SD) it is good if you write like 130.5 ±0.4

• We appreciate the comment, however, the "mean (SD)" is written in accordance with the guideline of PLoS ONE which states: "Properties of distribution. It should be clear from the text which measures of variance (standard deviation, standard error of the mean, confidence interval) and central tendency (mean, median) are being presented."

Discussion

Line 476-478---Why you raised this issue? Because as you told us this was not assed in your study. So it is good if you delete it.

• Thank you, we have deleted it.

Strength

Line 509---judgment of causality?? Your study design is cross-sectional how it could be???

As my perception it is good you minimize or summarize it and make your strength & limitation to one paragraph

• We also accepted this comment and minimized the size.

Conclusion

Line 536—you write modifiable risk factors but it is good if you indicate this statement above (line 529) when you listing factors associated with blood pressure. Because this is conclusion so it is good if you show us those risk factors you identified in your study as modifiable and non-modifiable.

• Thank you, we accepted your comment and moved the statement to the place where you suggested, and listed modifiable risk factors.

Reference

Line 596—reference 10—randomly I checked this reference and for this article there is authors so, it is good if you see and revise it.

• Thank you so much for the comment, we have revised all the references and made corrections accordingly. 

7. PLOS authors have the option to publish the peer review history of their article (what does this mean?). If published, this will include your full peer review and any attached files.

Do you want your identity to be public for this peer review? For information about this choice, including consent withdrawal, please see our Privacy Policy.

Reviewer #1: Yes: Ishag Adam

Reviewer #2: No

• No

---

## [Editor Report · Decision Letter 2]

27 Aug 2021

PONE-D-21-02507R2

Hypertension and blood pressure in southern Ethiopia: a community-based cross-sectional study

PLOS ONE

Dear Dr. Kumma:

Thank you for submitting your manuscript to PLOS ONE. After careful consideration, we feel that it has merit but does not fully meet PLOS ONE’s publication criteria as it currently stands. Therefore, we invite you to submit a revised version of the manuscript that addresses the points raised during the review process.

You please should make all the changes in the tables and analysis suggested.

We look forward to receiving your revised manuscript.

Kind regards,

Rosely Sichieri

Academic Editor

PLOS ONE

Journal Requirements:

Additional Editor Comments (if provided):

I am sorry to say that my understand is that this revised version still requires many changes.

The paper is too long, and it lacks focus.

1- There is no need of looking at levels of blood pressure and prevalence. Authors should keep only prevalence analysis. All diastolic and systolic analysis should be deleted.

2- Title should be changed - prevalence of hypertension in adults of xxxx and related factors.

3- The aim as written is to compare urban and rural areas??? This is not true.

The aim is to determine prevalence of hypertension and pre-hypertension and related factors.

4-multilevel analysis should be explained/ Was it used to account for the complex design? Are weighting included??? The ICC data is not necessary to show the intracluster? Complex surveys have specific programs called survey in SAS and Stata. They should be used.

4- Keep only the factors related to hypertension in the introduction and adjust for those factors that make sense. Multivariate analysis adjusting all for all make no sense.

5- Tables are wrong. Prevalences in the new tables by each factor should be showed. Not percentages in the column.

6- No need of table 1. Make the first the prevalence of hypertension and pre-hypertension by characteristics of the population. Weighted prevalence.

7- 7- the same for tables 2 and 3. All should be combined in one also with the prevalence by factors. Excluded all lipids and family store.

8- No need of table 4 and 5.

9- Review all the results and discussion after new analysis
---

## [Author Response · Author response to Decision Letter 2]

18 Oct 2021

Dear Dr. Rosely Sichieri,

We would like to thank you for your suggestions, and comments. We have worked on the comments provided by you. We have uploaded the following items during our resubmission: Responses to the editor, Revised Manuscript with Track Changes, and Revised Paper without Track Changes (Manuscript). Please see below our one-by-one response and explanations to questions and comments. Our responses and explanations to the questions and comments are indicated in bullet points.

Thank you for submitting your manuscript to PLOS ONE. After careful consideration, we feel that it has merit but does not fully meet PLOS ONE’s publication criteria as it currently stands. Therefore, we invite you to submit a revised version of the manuscript that addresses the points raised during the review process.

You please should make all the changes in the tables and analysis suggested.

• Thank you so much, we have made the changes in the tables and analysis based upon your suggestion. 

• Thanks, we made our decisions based upon PLOS ONE's publication criteria. 

a. Please clarify the sources of funding (financial or material support) for your study. List the grants or organizations that supported your study, including funding received from your institution.

• South Ethiopia Network of Universities in Public Health (SENUPH) funded the research; which in turn is funded by the Norwegian Program for Capacity Development in Higher Education and Research for Development (NORHED), the grant number of this research was ETH-13-0025.

• Wolaita Sodo University provided logistics and technical support.

b. State what role the funders took on the study. If the funders had no role in your study, please state: “

• The funders had no role in study design, data collection, and analysis, decision to publish, or preparation of the manuscript.

• There is no author who received a salary from South Ethiopia Network of Universities in Public Health (SENUPH) or Norwegian Program for Capacity Development in Higher Education and Research for Development (NORHED). The principal author is the academic staff of Wolaita Sodo University, Ethiopia. Therefore, I receive a salary from the Wolaita Sodo University, Ethiopia. However, the university had no role in the study design, data collection and analysis, decision to publish, or preparation of the manuscript. Currently, I am on study leave for a PhD study. I am doing my PhD study in a joint PhD program at Hawassa University and the University of Bergen. And this study is part of my PhD work, where the Wolaita Sodo University has no direct role in this study.

d. If you did not receive any funding for this study, please state: I received funding and the financial information was given above. 

Additional Editor Comments (if provided):

I am sorry to say that my understanding is that this revised version still requires many changes.

The paper is too long, and it lacks focus.

• Thank you, we appreciate your point and worked based on your comment.

1- There is no need of looking at levels of blood pressure and prevalence. Authors should keep only prevalence analysis. All diastolic and systolic analyses should be deleted.

• Thanks, we worked according to your suggestion. Therefore, we deleted diastolic and systolic analyses and analyzed only the prevalence of hypertension. 

2- Title should be changed - prevalence of hypertension in adults of xxxx and related factors.

• We would also like to appreciate your comment, we have now revised the title, and wrote based upon your suggestion. 

3- The aim as written is to compare urban and rural areas??? This is not true.

The aim is to determine the prevalence of hypertension and prehypertension and related factors.

• Thank you, we have also restated the aim of the study, according to your suggestion.

4- Multilevel analysis should be explained/ Was it used to account for the complex design? Are weighting included??? The ICC data is not necessary to show the intracluster? Complex surveys have specific programs called survey in SAS and Stata. They should be used.

• We accepted the comment and carried out the multivariate analysis using survey data analysis in STATA meaning after declaring the data set as survey data. 

5- Keep only the factors related to hypertension in the introduction and adjust for those factors that make sense. Multivariate analysis adjusting all for all makes no sense.

• Thank you, in the introduction, we listed factors with public health importance from studies conducted in Ethiopia and elsewhere that showed association with hypertension, and from a practical point of view, we kept or adjusted for those factors in the multivariate analysis. 

6- Tables are wrong. Prevalences in the new tables by each factor should be showed. Not percentages in the column.

• Thank you, we accepted the comment and worked accordingly. 

7- No need of table 1. Make the first the prevalence of hypertension and prehypertension by characteristics of the population. Weighted prevalence.

• Thank you, we made the first table prevalence of hypertension and prehypertension by characteristics of the population as you suggested, and also carried out the weighted prevalence. 

8- The same for tables 2 and 3. All should be combined in one also with the prevalence by factors. Excluded all lipids and family store.

• Thank you, we also made the second table as suggested above with the prevalence by the behavioral and other factors, excluding lipids and family history.

9- No need of table 4 and 5.

• Thank you, we have omitted tables 4 and 5. We have now only two descriptive tables.

10- Review all the results and discussion after new analysis

• We also appreciate all your comments. We wrote the results and discussion based upon the new analysis.

---

## [Editor Report · Decision Letter 3]

10 Nov 2021

Prevalence of hypertension, and related factors among adults in Wolaita, southern Ethiopia: a community-based cross-sectional study

PONE-D-21-02507R3

Dear Dr. Kumma,

We’re pleased to inform you that your manuscript has been judged scientifically suitable for publication and will be formally accepted for publication once it meets all outstanding technical requirements.

Kind regards,

Rosely Sichieri

Academic Editor

PLOS ONE

Additional Editor Comments (optional):

Thanks for accepting all the suggestions.

Please, in the final version exclude the last column in table 3 (p-value) not needed. The CI has the same meaning. Also in this table change No- for number of participants for N.
---

## [Editor Report · Acceptance letter]

6 Dec 2021

PONE-D-21-02507R3 

Prevalence of hypertension, and related factors among adults in Wolaita, southern Ethiopia: a community-based cross-sectional study 

Dear Dr. Kumma:

I'm pleased to inform you that your manuscript has been deemed suitable for publication in PLOS ONE. Congratulations! Your manuscript is now with our production department. 

Kind regards, 

on behalf of

Dr. Rosely Sichieri 

Academic Editor

PLOS ONE